# Use of Decision Trees to Evaluate the Impact of a Holistic Music Educational Approach on Children with Special Needs

**Liza Lee** [1,2] **and Ying-Sing Liu** [2,*]

1   Department of Early Childhood Development and Education, Taichung 41349, Taiwan; lylee@gm.cyut.edu.tw
2   College of Humanities and Social Sciences, Chaoyang University of Technology, Taichung 41349, Taiwan
*   Correspondence: liuyingsing@yahoo.com.tw

**Abstract:** In this study, decision trees were used to develop a pre-assessment model to help ascertain the impact of music education on children with special needs. The focus of the study was the application of an educational curriculum for 16 weeks, five sessions of 40 min duration per week, using the Holistic Music Educational Approach for Young Children (HMEAYC). The pilot program was implemented with children with special needs to measure its learning effectiveness. The methodology proved a better indicator for improved learning and a better measure of learning effectiveness. Statistical tests confirmed significant improvements in the values of the learning evaluation indices measured by HMEAYC after its implementation in children with special needs, supporting the positive effect of the implementation of HMEAYC for Taiwan's special needs young children. For children with better learning results, the accuracy of the decision tree model was 84.0% for in-sample and the sensitivity equaled 98.0%. The results support the future development of evaluation models through machine learning languages, pre-assessment of the effectiveness of the implementation of HMEAYC, and the use of continuous investment in educational resources to improve the efficiency of special early childhood education in resource consumption for sustainable development.

**Keywords:** special early childhood education; HMEAYC; data mining; pre-assessment learning effectiveness; sustainable development

## 1. Introduction

Past studies have confirmed that children with special needs are significantly affected in terms of learning, language, and behavior [1,2]. However, music experience is more important than what the eye sees [3] and shows the importance of music to people. Many studies have supported the use of music to influence the grammar and semantic processing of young children [4–6]. In particular, studies of children with special needs have found that music-impaired children can improve their language skills through music activities [7], so the use of music as education for children with special needs has received attention.

The Holistic Music Educational Approach for Young Children (HMEAYC) [8–11] is a form of music education that integrates local culture and uses computer technology and equipment to develop music education courses for preschoolers. This course has long been extended to music education for young children in general [10,12], and in recent years, it has been developed for children with special needs [9,12] and has been used in early therapy music learning courses for special young children, showing a significant positive impact on language and communication, body movements, attention, emotional stability, and interpersonal interaction [8–10,12].

In the past, research involving the application of data mining techniques has successfully explored sustainability education and development issues [13,14]. This study refers to the data mining techniques developed in the past and uses decision trees to pre-assess the learning effectiveness of the HMEAYC in early childhood education for children with special needs in Taiwan. This assessment model was developed to measure the effectiveness

of the HMEAYC when implemented in special early childhood music education, given the high social resources and costs involved. It uses the decision tree of machine learning language, and it is expected that in the future, as the evaluation data continues to expand, it will be possible to develop a dynamic system that can accurately evaluate participants' learning effectiveness in advance. Finally, through the established assessment system, the factors within the HMEAYC that are achieving better learning results for children with special needs can be assessed in advance for music educators as a reference, to improve efficiency in the use of resources in special education, to maintain sustainable development.

## 2. Data and Empirical Method

### 2.1. Data and Variables

Participants in this study were preschool children with special needs enrolled in nonprofit, early treatment institutions who were identified by Taiwan's local government as eligible and were aged between 24 and 60 months. The data were collected between 2012 and 2019. A total of 81 young children with special needs were issued with a certificate of physical and mental disability by the local government and were eligible for the study. Participants were trained for 16 weeks, five days a week, once a day for 40 min, using the HMEAYC [12]. During the implementation of the education course, the data were measured by nonparticipant observation and then cross-compared by more than two professional observers and converted into valid data.

The independent variables were as follows: (1) Sex: when "boy" given 1 and "girl" is 2; (2) age: 24 months < age $\leq$ 72 months; (3) disability categories: mental retardation/intellectual disability was given 1, visual impairment was given 2, hearing impairment was given 3, language impairment was given 4, physical handicaps were given 5, and cerebral palsy was given 6. Finally, the seven learning assessment indicators were divided into language comprehension (code: LC), language expressiveness (code: L), self-control (code: SC), self-directed (code: D), interpersonal relationships (code: R), social skills (code: S), and physical movement (code: P) [12]. The learning assessment index measures the pre-measured values (code: pre), the mid-term values (code: mid) for the full eight weeks, and the post-measured values (code: post) after the implementation of the full 16 weeks. The change rate of the learning assessment indicator during the period after 8 weeks of implementation of the HMEAYC is $X_{PM} = (X_{mid, i} - X_{post, i})/X_{post, i}$. Similarly, it is possible to calculate the rate of change of the learning assessment indicator at the end of the period after 16 weeks of implementation of the HMEAYC.

The target variable (dependent variable) is defined as the learning improvement ability indicator $IAI(n) = \begin{cases} 1, \ n \geq 3 \\ 0, \ \text{otherwise} \end{cases}$ ; where $IIAI(n)$ measures the improvement after 16 weeks of implementation of the HMEAYC in the seven assessment indicators for preschool children with special needs, and $n$ is the number of indicators with a post-measured value greater than the pre-measured value. If the number of indicators with a post-measured value more significant than the pre-measured value is at least three ($n \geq 3$), $IIAI(n) = 1$; otherwise, $IIAI(n) = 0$.

### 2.2. Decision Trees

Data mining is a technique that combines algorithms, artificial intelligence, and machine learning and uses databases to explore specific rules for analyses from a large amount of data, to find unknown information that is needed to resolve the issue. Decision trees are a common and reliable method in data mining technology [15] and have been used in the field of education [16]. This study used classification and regression trees (CART) [17,18]. The Gini index was used as a branching criterion (splitting criteria) in the CART to identify branching variables, tree width, and depth to establish a binomial

classification of the decision tree structure. The Gini index is a measure of the data set S for all categories of impurity, and it is defined as follows:

$$G(S) = 1 - \sum_{i=1}^{k} p_i^2. \tag{1}$$

If the candidate's branch variable, $v$, has $h$ properties that can be separated into $v_j$, in the case of a jth property, the number that appears is $N_j$ and the probability is $p_j$. When the property value is $v_j$, where j = 1,2,3, ... , $h$, the impurity index obtained is $G(v_j)$. The total data impurity in the candidate branch variable, $v$, will be equal to the property value under all $v_j$, with expected impurity given by $G_v(S) = \sum_{j=1}^{h} p_j \times G(v_j)$. When $G_v(S)$ is smaller, the properties separated by the variable are more appropriate as branching variables, and a comparison of the impurity values of each candidate branch variable can identify the most suitable branching variable. In general, the branching variable of impurity is used as the basis for the segmentation of the parent node by the greatest reduction. There is a basis for measuring changes in impurity as follows:

$$\Delta G(v) = G(S) - G_v(S). \tag{2}$$

When $\Delta G(v) \geq 0$, node N does not consider branching. If $\Delta G(v) < 0$, node N considers branching several child nodes.

### 3. Results

There were 81 participants in the study, 53 boys (65.4%) and 28 girls (34.6%), aged between 28 and 71 months (average $\pm$ standard deviation was 47.54 $\pm$ 14.31). Table 1 shows the measured indices for the descriptive statistics. The post-measured average values of the seven learning assessment indicators are higher than the pre-measured measures, while the mean of change in Mid. and change in Post. is positive, showing that the average of the seven learning assessment indicators improved after the implementation of the HMEAYC for the full 16 weeks (full eight weeks).

**Table 1.** The measured indices for the descriptive statistics ($N$ = 81).

| Index | Pre. | Mid. | Post. | Change in Pre.-to-Mid. | Change in Pre.-to-Post. |
|---|---|---|---|---|---|
| | Mean $\pm$ S.D. | Mean $\pm$ S.D. | Mean $\pm$ S.D. | Mean $\pm$ S.D. | Mean $\pm$ S.D. |
| LC | 6.346 $\pm$ 3.947 | 7.383 $\pm$ 4.051 | 6.926 $\pm$ 4.071 | 0.247 $\pm$ 0.289 | 0.129 $\pm$ 0.202 |
| L | 10.185 $\pm$ 4.413 | 10.556 $\pm$ 4.626 | 10.877 $\pm$ 4.559 | 0.049 $\pm$ 0.153 | 0.117 $\pm$ 0.368 |
| SC | 7.200 $\pm$ 4.202 | 7.450 $\pm$ 4.134 | 7.457 $\pm$ 4.237 | 0.065 $\pm$ 0.168 | 0.078 $\pm$ 0.223 |
| D | 11.025 $\pm$ 7.381 | 11.407 $\pm$ 7.448 | 11.728 $\pm$ 7.589 | 0.047 $\pm$ 0.130 | 0.100 $\pm$ 0.229 |
| R | 3.911 $\pm$ 2.737 | 4.111 $\pm$ 2.734 | 4.481 $\pm$ 2.802 | 0.151 $\pm$ 0.306 | 0.281 $\pm$ 0.454 |
| S | 15.753 $\pm$ 6.780 | 16.012 $\pm$ 6.600 | 16.444 $\pm$ 6.620 | 0.049 $\pm$ 0.155 | 0.120 $\pm$ 0.473 |
| P | 8.346 $\pm$ 6.948 | 8.593 $\pm$ 7.016 | 8.914 $\pm$ 7.145 | 0.040 $\pm$ 0.092 | 0.094 $\pm$ 0.140 |

Table 2 shows the results of the $t$-statistics [19] and Wilcoxon signed-rank tests [20] on the paired difference between the pre- and post-measured values based on the seven learning evaluation indicators for special needs young children after 16 weeks of implementation of the HMEAYC. These two statistical methods are traditionally used to determine whether there is a significant difference between the before- and after-training measurement indicators, to prove whether there is learning effectiveness. The results determined by the $t$-statistics, at a significant level of 5%, show that the average of the paired differences is significantly positive ($p$-values < 0.05) for the seven learning evaluation indicators, showing that the implementation of the HMEAYC for special needs young children had a significant positive effect in all indicators. The results from the Wilcoxon signed-rank tests were found to be the same as for the $t$-statistical tests.

**Table 2.** The tests of the differences between the paired samples (*N* = 81).

| Paired Sample (Post–Pre) | Mean (Post–Pre) | *p*-Value [#] (*t* Stat.) | Wilcoxon Matched-Pairs Signed-Rank Tests | |
|---|---|---|---|---|
| | | | *z* Stat. | *p* Value [#] |
| LC | 0.580 ** | 0.000 | −5.716 [b] | 0.000 |
| L | 0.691 ** | 0.000 | −4.703 [b] | 0.000 |
| SC | 0.325 ** | 0.003 | −3.169 [b] | 0.002 |
| D | 0.704 ** | 0.000 | −5.375 [b] | 0.000 |
| R | 0.646 ** | 0.000 | −5.739 [b] | 0.000 |
| S | 0.691 ** | 0.000 | −4.839 [b] | 0.000 |
| P | 0.568 ** | 0.000 | −5.729 [b] | 0.000 |

Note: ** represent significance at the 1% levels; [b] is based on the negative-ranked; [#] is based on two-tailed.

Figure 1 shows the decision tree model established by using the CRT growth method. The target variables are $IIAI(X)$, which contain Ppre, disability categories, SCPM (PM: change in Pre.-to-Mid.), Spre, LCpre, Dpre, SCpre, Lpre, Rpre, SPM, LCPM, DPM, PPM, RPM, LPM, age, and sex. The minimum number of observations in the parent node was 10, the minimum number of observations in the child node was five, the tree depth was four, the number of nodes was 11, the number of endpoints was six, and the significance level was 0.05. In this study, the risk value of the obtained pattern was 0.284, the standard error was 0.050, the sensitivity was 98.0%, and the accuracy was 84.0%, where $IIAI(X) = 1$ (the prior probability was 0.617). The importance of the variables is sequenced as LCPM (0.094; regularization 100.0%), Ppre (0.086; regularization 91.8%), disability categories (0.061; formalization 65.0%), DPM (0.060; regularization 63.5%), Rpre (0.059; normalization 62.5%), SCpre (0.052; normalization 55.0%), LCpre (0.043; normalization 46.1%), SCPM (0.038; normalization 40.4%), and RPM (0.035; normalization 37.5%).

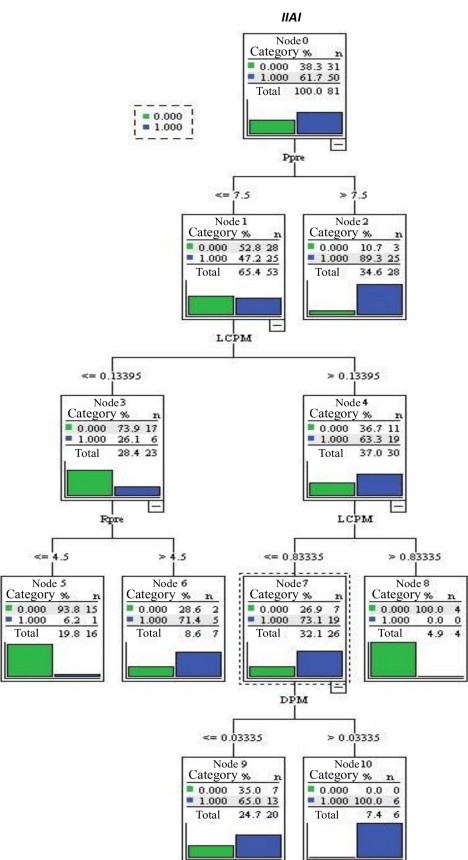

**Figure 1.** A tree structure pattern for the Holistic Music Educational Approach for Young Children (HMEAYC).

## 4. Conclusions

After the implementation of HMEAYC course training for young children with special needs for 16 weeks, five sessions of 40 min duration per week, the participants showed a significant improvement in the measured learning assessment indicators, supporting the implementation of the HMEAYC for young children with special needs. In addition, the decision tree model, established to evaluate the implementation of the HMEAYC, achieved better learning results (at least three of the seven indicators have a score increase), with a sensitivity of 98.0% and an accuracy of 84.0%. The first layer in the tree structure to implement the HMEAYC with better learning results contained the pre-measured physical movement value, which had a higher proportion (62.4%) when the pre-measured value of physical movement was $\geq 7.5$. Second, three factors, namely, the pre-measured values of the interpersonal relationship and mid-term changes in language comprehension and self-directed, will be factors in assessing better learning performance. Finally, this model is an earlier assessment of the implementation of the HMEAYC learning effectiveness, from the established tree model. In the future, with the continuous expansion of the data, it is expected that this model will have a relatively good predictive capability and can be used as a pre-assessment tool for the implementation of HMEAYC course training. Using this approach to identify whether or not children with special needs have invested or continue to invest in educational resources will improve the consumption of special education resources and enable the efficient use of educational resources for the sustainable development of special education.

**Author Contributions:** Conceptualization, L.L.; methodology, Y.-S.L.; software, Y.-S.L.; validation, L.L.; formal analysis, L.L.; investigation, L.L.; resources, L.L.; data curation, L.L.; writing—original draft preparation, L.L.; writing—review and editing, L.L.; visualization, L.L.; supervision, L.L.; project administration, L.L.; funding acquisition, L.L. Both authors have read and agreed to the published version of the manuscript.

**Funding:** This research received no external funding.

**Institutional Review Board Statement:** This study exclude did not require ethical approval, and not applicable for studies not involving humans or animals.

**Informed Consent Statement:** Informed consent was obtained from all subjects involved in the study.

**Data Availability Statement:** Data sharing is not applicable to this article.

**Conflicts of Interest:** The authors declare no conflict of interest.

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
