# Peer review of "Use of Decision Trees to Evaluate the Impact of a Holistic Music Educational Approach on Children with Special Needs"

_sustainability, doi:10.3390/su13031410_

Round 1

Reviewer 1 Report

Thank you for submitting your manuscript. There is a need for research in special music education focused in early childhood and I know Taiwan is one of the leaders in early childhood education, and we appreciate your expertise. 

That said, I really do not understand this paper. Part of that is because I am a qualitative researcher and the statistics and results have no meaning to me. But, I should be able to understand the study if it is explained so qualitative researchers can make sense of the results. Perhaps there should be an additional section that clearly explains what the intervention was, how it impacted the children and another section labeled "Implications for Music Education" that helps someone who needs to know if there are suggestions for further research, or problems in this study that should be further examined. 

Terms need to be explained, for example, what is Holistic Music Educational Approach? Is that a formal approach in early childhood music? If it is, I don't know about it. If it isn't a formal approach, it should not be capitalized. Other terms that confused me are "music-impaired", Holistic education" in #39, it sounds like an institute? Also, I am unfamiliar about decision trees. What is that? 

What was the intervention/implementation of HMEAYC? What do you do and how does that impact language? I never understood how you demonstrated it through your statistical work. Are children with a variety of disabilities being compared with each other? Do they all have speech/language disorders? How are these children diagnosed? Does a speech/language therapist do that or are the music education/therapy professionals determining disorders? 

I know if I understood what you did it would likely be a meaningful piece of research, but as it stands, I could never use this as a reference because I don't understand how it fits into the research literature. 

The article needs to be revised for clarity taking into consideration that not all readers understand your statistical instruments. The article also needs expansion so that the reader understands what the treatment was and how and why children responded the way that they did. Adding a conclusion or summary plus an implications for music education section would help the reader to follow your conclusion. Finally, an editor for cleaning up English grammar mistakes should be used. If one is not available, consider downloading the application Pearson Writer. It is a very good electronic editing tool that goes beyond what Microsoft Word is able to edit. 

Author Response

Thank you for Reviewer 1 comments on our manuscript, while giving the author to clarify and add to the content of this article, as well as our responses in additional  files.

Reviewer 2 Report

You have written an interesting paper with very positive outcomes. The description of the content is on the short side. I am particularly curious to the conducted intervention that has been used. However, that is not the focus of the paper.

Author Response

Thank you for Reviewer 2 comments, and our responses in additional files.

Round 2

Reviewer 1 Report

Thanks for spending the time to revise the paper. I understand it now. I would still like to understand what it is that you specifically did using the HMEAYC. If I were to have a colleague interested in early childhood music education research, I'm not sure they could read this paper and know what you did. However, what you did with young children clearly works!